# Strengthening Food Systems Governance to Achieve Multiple Objectives: A Comparative Instrumentation Analysis of Food Systems Policies in Vanuatu and the Solomon Islands

Erica Reeve [1,2,*], Amerita Ravuvu [3], Anna Farmery [4], Senoveva Mauli [4], Dorah Wilson [4], Ellen Johnson [1] and Anne-Marie Thow [1]

1   Menzies Centre for Health Policy, Charles Perkins Centre (D17), School of Public Health, University of Sydney, Sydney, NSW 2006, Australia; ellen.johnson@sydney.edu.au (E.J.); annemarie.thow@sydney.edu.au (A.-M.T.)
2   GLOBE, Institute for Health Transformation, Faculty of Health, Deakin University, Geelong, VIC 3220, Australia
3   Non-Communicable Disease (NCD) Prevention and Control Programme, Public Health Division, Pacific Community (SPC), Suva, Fiji; ameritar@spc.int
4   Australian National Centre for Ocean Resources and Security, University of Wollongong, Wollongong, NSW 2500, Australia; afarmery@uow.edu.au (A.F.); sm630@uowmail.edu.au (S.M.); deejwils@gmail.com (D.W.)
*   Correspondence: e.reeve@deakin.edu.au or erica.reeve@sydney.edu.au
**Abstract:** Political leaders from around the world are demonstrating interest in adopting food policies that account for the economic, health, social and environmental dimensions of food. In the Pacific Islands, decades of experience in implementing multisectoral NCD and climate policy has indicated that operationalising food systems policies will be challenging. We aimed to identify opportunities for food systems sectors to more strongly promote nutrition and environmental sustainability in addition to economic objectives. We conducted a comparative documentary analysis of 37 food systems sector policies in Vanuatu and the Solomon Islands. We applied theories of agenda-setting to examine how the frames employed by different sectors, and evident in policy content, shaped policy priorities and activities. We identified a predominately economic framing of issues affecting food systems sectors. Though there were clear policy aims to produce enough food to meet population dietary requirements and to promote an environmentally resilient food supply, aims operationalised more predominately through policy content were those that increase the contribution of productive sectors to food exports and import substitution. Food systems sectors in the Pacific Islands have clear aims to promote nutritious and environmentally resilient food systems, but policy instruments could more strongly reflect these aims.

**Keywords:** food systems policy; policy coherence; healthy and sustainable food; Pacific Islands; policy instruments

## 1. Introduction

Food systems are of critical importance to both natural environments and human health. Globally, concerns have been rising over the shift towards greater production and consumption of processed foods high in calories, salt, sugar and fat [1,2]. These foods have played a key role in the development of noncommunicable diseases (NCDs) [3–7], collectively responsible for over 71% of global mortality [8], whilst malnutrition in all its forms remain largely unaddressed [9]. The production, trade and consumption of food is a major contributor to climate change, environmental degradation and biodiversity loss [10,11]. Food systems play a critical role in economic and social development, but almost half of the world's population cannot afford a healthy diet [9,12], a problem that has been exacerbated by pandemic-induced economic downturn [9]. Additionally, agricultural systems more often favour the production and distribution of high-energy staple foods

and processed foods over fruit and vegetables, leading to unprecedented environmental damage [11]. At the recent UN Food Systems Summit (2021), governments around the world committed to balancing food production with health objectives, environmental sustainability and climate action through 'systems-wide change' [13]. It was agreed that all aspects of the food system need reorientation such that they promote health and wellbeing, restore and protect nature, promote livelihoods and inclusive economies, and that food systems need to be more responsive to local tradition and circumstance [13].

A critical next step is for countries to implement food systems policies that take into account economic, health and environmental dimensions of food, in an integrated way [13]. To be effective, food systems governance requires the engagement of policy actors from across a large number of sectors, including those that govern food production, distribution, transport, trade, processing, marketing and retail [14]. However, policy issues relating to these multiple dimensions are usually led by their respective sectors, for example, the health or environmental sector. Decades of experience in implementing climate change and NCD prevention policies has demonstrated that operationalising multilateral and cross-jurisdictional policy is challenging, because sectoral mandates and budgets can be narrow, and each sector is governed by a different set of policy objectives [15–19]. These factors undermine governments' ability to guide food systems so that they are efficient food producers but also minimise negative health and environmental outcomes [20]. In addition, government action is dependent on political priority, and there is a perceived lack of priority for food and nutrition policy among political leaders and non-health government actors [16], and difficulties mobilising and sustaining their commitment. Environmental concerns have likewise been met with governance challenges as governments tend to absorb them into policy in an ad hoc way [21]. Strengthening policy coherence and reducing policy fragmentation are, thus, critical elements of global recommendations for action on food systems [22]. However, addressing policy fragmentation requires a major review of governance structures, and there is a need to demonstrate for countries more practical ways to systematically improve coherence of food systems policy across economic, health and environmental dimensions of food.

This paper aims to identify opportunities for food systems policy in two Pacific Island Countries to simultaneously promote multiple dimensions of food systems, particularly nutrition and environmental sustainability dimensions. The Pacific Islands make a useful backdrop for this study because of their significant contribution to global food systems via ocean resources and other high value products [23]. As in many other low- and middle-income countries (LMICs), rapidly transitioning food systems have led to Pacific Islanders experiencing high rates of diet-related NCDs [24], childhood obesity [25] and micronutrient deficiencies [26,27]. Pacific Island Countries also report shortfalls in food production that are exacerbated by climate change and resource exploitation [23,28]. In response to this shortfall, and their participation in the UN Food Systems Summit, Pacific Island leaders have developed a set of policy priorities for transforming food systems. These priorities have included drawing on traditional knowledge and expertise for guidance on agro-ecological solutions, strengthening governance of ocean and coastal marine resources given their significance to global food supply, and reorienting trade systems to promote health and environmental outcomes [23]. In this paper, we examine food systems policy in the Solomon Islands and Vanuatu, where sociodemographic changes have compromised traditionally subsistence-based lifestyles and led to increased consumption of processed foods high in sodium, sugar and hydrogenated fats. Like many other LMICs, governments there are now faced with the task of ensuring food systems policy is consistent with sustainable development objectives, and that policies are adopted and implemented by stakeholder groups with vastly different objectives. In this study, we use an instrumentation approach to analyse current food systems policies, with a focus on coherence between policy problems, and the food systems policy instruments used to address them. We then use this analysis to reflect on opportunities for nutrition and environmental sustainability

aims to be more strongly operationalised through food systems policies in other LMICs, with reference to global recommendations.

## 2. Materials and Methods

### 2.1. Study Design

We conducted a comparative documentary policy content analysis, focused on policy instrumentation. Policy content analysis is the systematic description of policy content to convey the meaning behind its content [29]. This policy content analysis focused on coherence between policy aims and objectives, and the policy instruments used to respond to these objectives [30]. An instrumentation approach to policy analysis can provide insights into the degree of political commitment to an issue, as expressed through general priorities (rhetoric and framing) or in concrete commitments [31–33]. It also facilitates differentiation on the degree of priority and commitment, and has been used previously to assess how policies are utilised to address different priorities [34,35]. For the purposes of this study, and based on food systems frameworks [36], 'food system policies' are defined as policy documents (strategies, plans of action, etc.) in the following sectors: Agriculture, Livestock, Fisheries, Infrastructure, Industry, Trade and Investment, and Finance. Agriculture, Fisheries and Livestock are considered the 'productive sectors'. Policies from the health and environmental sectors were not a focus of the study. Legislation across all relevant domains was considered cross-governmental.

### 2.2. Conceptual Framework for the Study

The data collection and analysis drew on political science frameworks for agenda setting and policy instrumentation. Food systems policy involves nearly all government sectors, in which policy makers take action with often limited knowledge resources and underlying sectoral biases [37,38]. Theories of agenda setting highlight the importance of the frames employed by different sectors, and evident in policy content, in shaping policy priorities and activities [39–41]. In a policy context, 'framing' refers to the ways in which policy makers understand and define a policy issue or problem [42]. Frames evident in policy content can, thus, provide insights into sectoral understandings of policy problems or concerns and policy priorities [43]. Policy concerns represent value judgements regarding societal 'problems' that should be addressed by policymakers through (certain) policy 'solutions' [44].

Policy instruments are the tools used by governments to address policy concerns, and to bring public and institutional behaviours into alignment with policy goals [45–47]. Governments use policy instruments to establish expected courses of actions for institutions and individuals. Of significance is that policy instruments are not neutral, they produce varying effects based on their mechanisms for social control or fostering relevant behaviours [31]. In particular, they vary by the degree of authority, coercion or power to incite motivation [48]. In this way, an instrumentation lens to policy study provides critical insight as to the level of importance policymakers place on an issue.

Instrumentation typologies vary conceptually based on the degree of coercion, the focus of governing resources or the behaviour motivations [49]. In this paper, we borrow constructs from the instrument typologies of Vedung (regulation, economic, information) [48] and Ingram (authoritative, incentives, capacity) [50] to develop a typology that differentiates instruments depending on the degree of coercion (Table 1). We also disaggregated Ingram's definition of incentives into 'direct' from 'indirect' incentives to differentiate between those mechanisms that directly facilitate action (e.g., provision of farming inputs or sanctions), compared with those that act more indirectly by reducing barriers to participation (e.g., veterinary services), or offering strong motivation (e.g., tax rebates) for action.

**Table 1.** Proposed instrument typology for discussion.

| | Instrument Typology | Definitions/Origin | Included Instruments |
|---|---|---|---|
| 'Harder' | Authoritative | Regulatory and organisation structures with high degree of coercion and no tangible incentive for action apart from civic loyalty [50]. | Legislation, regulation, workplace compliance, policy, zoning |
| | Incentives-Direct | Instruments that manipulate benefits and costs to create tangible payoffs (positive or negative) to induce action or extinguish activities [50]. These assume individuals have opportunity to take action, recognise this opportunity, and have sufficient capacity to take action [50]. | Grants, business incubation Sanctions Equipment for farming and processing Taxes, charges Agricultural inputs Land allocation |
| | Incentives- Indirect | Instruments that manipulate benefits and costs (positively or negatively) to motivate action [50], including the provision of services that remove barriers to participation and connections to other actors for mutual benefit (own). These assume individuals have opportunity to take action, recognise this opportunity, and have sufficient capacity to take action [50]. | Quality assurance services including weight calibration, soil High value services including veterinary and machinery repair Loans and Loan guarantees General shared infrastructure important to food system (e.g., feed mills, market storage) Tax concessions or rebates Formation of farming cooperatives and producer associations to share inputs |
| | Knowledge and skills building | Instruments providing guidance, training and education that enable people to carry out an activity. These assume that incentives and motivations are in place, and the only barriers remaining are information or skill-related. These include social and organisational resources and support [50], and softer policies including guidelines and voluntary standards [48]. | Knowledge and capacity building Awards and certificates Technical guidance |
| 'Softer' | Infrastructure [a] | Improvements to general infrastructure that are necessary to the functioning of food systems | Waste management, water ICT, telecommunications Roads, transport Land planning and allocation |

[a] Authors own definition.

*2.3. Data*

This documentary policy analysis drew on data from current policies and legislation of government sectors relevant to food systems governance in Vanuatu and the Solomon Islands, covering years ranging from 2013–2031. We defined sectors relevant to food systems using a the food system framework provided by the High Level Panel of Experts on Food Security and Nutrition [36]; these included agriculture, trade, commerce, industry, infrastructure, fisheries and finance.

We used government organisational structures to guide searches for legislation, strategic or corporate plans and policies across government websites. Missing documents were sought through collaborators working in each country. NVIVO™ was used as a database to store and organise policy documents for extraction against a predetermined coding framework [36].

*2.4. Expert Review*

This work was overseen by the Pacific Food Policy Project Advisory group. The Advisory Group is comprised of senior policy experts in a range of key health, environment,

and academic agencies and community groups relevant to food systems in Vanuatu and the Solomon Islands, together with representatives from regional agencies, including FAO, the Pacific Community (SPC), and the research team. Experiences of the Advisory Group shared at quarterly meetings informed the development of the research concept, and three members of the group approved the analytical approach to ensure that it generated useful meaningful outcomes. An expert group member from each country validated the included documents and directed us to additional resources, as well as reviewing the findings to maintain relevance for policymakers.

*2.5. Analysis*

We coded all documents using NVIVO™, using pre-determined codes based on our study frameworks (Table 2). Two reviewers finalised the codebook and definitions after separately coding and comparing the first three documents. The deductive coding framework enabled us to extract policy content on: food systems concerns and frames, agency objectives and priorities, and policy instruments, as per our study aims.

**Table 2.** Coding framework for analysis.

| Main Codes | Sub Codes |
| --- | --- |
| Framing | Environmental sustainability |
| | Problem(s)—food/food system |
| | Nutrition |
| Gender & youth | |
| Governance | Institutional strengthening |
| | Policy coordination |
| | Own agency's role (related food) |
| Policy implementation | Resourcing |
| | Partnerships |
| | Training |
| Policy instruments, relevant to food system | |
| *Legislative or regulatory* | Legislative—(food) businesses |
| | Legislative—land & water (production) |
| | Legislative—trade, marketing & consumers |
| *Economic instruments* | Subsidies |
| | Tax policy |
| *Incentives* | Access to credit |
| | Incentives-consumption |
| | Incentives—food businesses |
| | Incentives—primary production |
| | Infrastructure (general, relevant to food eg transport) |
| *Knowledge, skills, training* | Knowledge & information—consumers |
| | Knowledge & information—food businesses |
| | Knowledge & information—primary production |
| Policy Monitoring & Evaluation | |
| Policy objectives | Overarching policy objectives |
| | Specific priority foods |
| Reference to other policies | National policies |
| | Regional and international policies |

We analysed the contents of each code and documented the contents of each code, noting any points of difference or similarities within or between each country. We first documented the food systems policy concerns and aims for each country, and the framing of three core food system aims (economic, nutritional and environmental) that were consistent across both. This provided us with an overview of the food system policy landscape and its broad intent. Second, we described, from the codes, the instruments used to address each of those policy aims, with reference to our instrumentation typology. We present below our findings against the leading policy concerns.

## 3. Results

### 3.1. Overview of Findings

We identified 37 policy documents relevant to food systems in the Solomon Islands and Vanuatu across 5 main sectors (Table 3). Presented below are the concerns ('policy problems') spanning economic, environmental and nutrition aspects of food systems, but differing by sector. Analysis of policy aims and instruments is then presented, with findings grouped by the following themes: economically-oriented aims; environment and natural resource-oriented aims; food security and nutrition-oriented aims.

**Table 3.** Policies included in the analysis.

| Food Systems Sector | Solomon Islands | In-Text Abbreviation | Vanuatu | In-Text Abbreviation |
|---|---|---|---|---|
| Agriculture and livestock | • Solomon Islands Agriculture Sector Growth and Investment Plan 2021–2030 | Sol Agri | • Agriculture Sector Policy 2015–2030<br>• Vanuatu National Livestock Policy 2015–2030 | Van Agri<br>Van Live |
| Fisheries | • National Fisheries Policy 2019–2029 (Sol Fish) | Sol Fish | • Vanuatu National Fisheries Sector Policy 2016–2031 | Van Fish |
| Commerce and industries | • Ministry of Commerce, Industry and Labour Corporate Plan 2020–2024<br>• Micro, Small and Medium Enterprises (SMEs) Policy and Strategy | Sol Industry | • National Industrial Development Strategy 2018–2022<br>• Marketing Strategy and Business plan | Van Industry |
| Finance, trade and investment | • Corporate Plan (2020–2022)<br>• 2021 budget speech<br>• Trade Policy Framework (2015) | Sol Fin<br><br><br>Sol Trade | • 2021 budget policy statement<br>• Trade Policy Framework Update (2019–2025)<br>• National Investment Policy Statement | Van Fin<br><br>Van Trade<br><br><br>Van Food |

**Table 3.** *Cont.*

| Food Systems Sector | Solomon Islands | In-Text Abbreviation | Vanuatu | In-Text Abbreviation |
|---|---|---|---|---|
| Infrastructure and planning | • Corporate Plan (2016–2020)<br>• National infrastructure development plan (2013–2023)<br>• National Water Resources and Sanitation Policy (2017) | Sol Infra | • Corporate Plan (2018–2020)<br>• National Land Subdivision Policy (2019) | Van Infra |
| Legislation | • Environment Act 1998<br>• Consumer Protection Bill (1995)<br>• Measurements and Weights Act (1996)<br>• Fisheries Management Act (2015)<br>• Biosecurity Act (2013)<br>• Foreign Investment Act (2005)<br>• The Pure Food Act (1996)<br>• Price Control Act (1996)<br>• Planning and Development Act (1980) | | • Convention on Biological Diversity Act (2006)<br>• Environmental Management and Conservation Act (2006)<br>• Fisheries Act (2006)<br>• Industry Development Act (2014)<br>• Food (Control) Act 2006<br>• Foreign Investment Act (2019)<br>• Price control Act (1974)<br>• Water Resources Management Act (2006) | |

*3.2. Policy Aims and Instruments*

3.2.1. Key Concern 1: Food Systems Are a Substantial Economic Concern within the Agriculture, Industry and Trade Sectors

The food system was described as a significant cause of economic concern within policy documents of countries, particularly framed in view of the declining performance of agricultural production for export markets, including, for example, the 'low and erratic production' of export crops, including coffee, kava, cocoa, vanilla and spices (Sol Agri, Van Industry) (Table 4).

> "*Production and productivity of the agriculture sector is low which negatively affects food security, national self-sufficiency levels, export earnings, employment generation in agriculture and allied sectors, and rural livelihoods in general*" Sol Agri

Economic impacts—namely declining export production, import dependance, and the substantial proportion of food leaving both countries without value-adding or processing—associated with declining production were framed as missed opportunities (Van Industry, Van Agri, Sol Agri, Sol Fish).

> "*We have not maximised the potential of the fisheries sector because our planning for this sector in the past has not been adequate... our fish continue to be landed in other*

*countries, supporting the economies of those countries while our government struggles to generate revenue to support its services*" Van fish

"*Vanuatu's focus on primary production and processing captures only a fraction of the value-added potential of coconut. Coconuts in Vanuatu are currently used mainly for copra production and some copra oil production while the rest being wasted*". Van Industry

Policy documents cited key economic statistics to substantiate production concerns. In Vanuatu, production losses were reported as a key reason for a 31% decline in exported goods between 2014 and 2015, and a 29% increase in the value of imports. The Solomon Islands Agriculture Investment Plan 2021–2030 outlined that agricultural export values had declined from SBD 564 million to 375 million in the years between 2011 and 2018, with agriculture's share of export declining from 28% to 8% over that time. At the same time, agricultural imports had increased over 10 years to reach SBD 630 million by 2018 (Sol Agri), largely due to imports of rice, poultry, wheat, coffee, tea and spices. Trade policy documents attributed the worsening trade deficit to agricultural production losses (Van Trade).

**Table 4.** Main policy concerns and aims expressed in food systems policy.

| Food System Outcome Area | Dominant Frames | Key Concerns | Leading Policy Aims |
|---|---|---|---|
| Economic And Livelihood | Economic impacts/consequences of low production | • Reduced contribution of agriculture to livelihoods, exports and trade <br> • Missed opportunities <br> ○ import dependance <br> ○ minimal or no value-adding prior to export | 1. Increase the contribution of productive sectors for import substitution and export trade, and industry development to promote domestic value-adding |
| Environmental | Economic impacts/consequences of environmental exploitation, degradation and natural disasters | • Impacts of climate change on natural resources, exacerbating inadequacies in production <br> • Poor resource management and commercial exploitation, reducing economic return for future | 2. Promote an environmentally resilient food supply and reduce environmental exploitation |
| Nutrition | Food security in future periods of vulnerability | • Potentially widening gaps between production and nutrition requirements in the context of population growth <br> • Protein sufficiency | 3. Produce enough food to meet population requirements for healthy diets |

Agriculture sector policy documents identified a range of drivers for low production (in relation to it being a major economic policy concern); the majority of these were described as emanating at the 'farm' level. The drivers of low production included customary land ownership challenges (Van Agri, Sol Agri), poor land use planning (Sol Agri), and the allocation of land for non-agricultural development. Limited farmer knowledge around improved methods, mechanised and animal-assisted agriculture, animal breeding and animal feed formulation (Van Agri, Van Livestock, Sol Agri) were expressed as being exacerbated by the absence of science-based advisory services, and difficulties communicating with farmers (Van Agri).

Other challenges, beyond the farm, included the high costs associated with accessing inputs for primary production and limited access to finance and credit by farmers linked to a lack of confidence by financial institutions to back agriculture (Van Agri, Sol Agri). Remoteness from markets and dispersed farming populations limit effective trading partnerships (Sol Agri, Van Agri) and are exacerbated by deteriorating transport infrastructure and high user-related expenses (e.g., boat transport, wharfage, storage) (Sol Infrastructure, Sol Trade). Women are rarely held as trustees in customary land ownership, which was identified as a barrier to their participation as business owners in the sector.

Food system policy concerns regarding production for export and import-substitution were attributed to the absence of comprehensive policy frameworks to facilitate production (Van Agri, Sol Agri) and trade (Sol Trade), the lack of coordination between producers and private and public partners (Van Agri), and the absence of commodity-specific value chain development plans between the agriculture sector and private sector. Food systems policy documents reflected that countries would need to build collaboration and trading partnerships through organised cooperatives, community governance and resource management, as well as by introducing more private sector investment in the sector.

3.2.2. Key Concern 2: Climate Change and Environmental Exploitation Presents a Risk for Productive Sectors

The vulnerability of food systems to the impacts of climate change, natural disasters and environmental exploitation was one of the predominating concerns evident in food systems policies across sectors. For example, climate change and disasters were listed as a leading contextual challenge in a large proportion of the documents reviewed. Policy concerns regarding the environment were ultimately framed as having economic consequences, including on livelihoods, agricultural output, trade opportunities and revenue (Van Agri, Sol Agri, Fish). Concerns included that climate change and environmental degradation have, and will continue to, severely exacerbate inadequacies in agricultural and fisheries production (Sol Fish, Van Agri, Sol Agri). The Solomon Islands Agriculture Sector Growth and Investment Plan (2021–2030) for instance, forecast reduced revenue associated with climate change and disasters and foresaw disaster-related impacts on future land development opportunities. The Vanuatu Agriculture Sector Policy 2015–2030 observed that tropical cyclone Harold had reduced exports of cocoa, beef and copra in Vanuatu.

*"Vanuatu's inability to increase and sustain agricultural production is exacerbated by the negative effects of climate change and climate variability."* Van Agri

*"Of the total damage and loss for the floods in 2014, 88 per cent is attributable to crops, 10 per cent to livestock, and 2 per cent to fisheries. The total effect to the sector amounts to USD 18.41 million, of which USD 1.50 million (8 per cent) is damage and USD 16.94 million (92 per cent) is loss. Extreme climate events and natural disasters are highly likely to significantly undermine agricultural productivity in the coming years."* Sols Agri

Productive sectors also flagged concern around the commercial exploitation of natural resources by export industries (specifically fisheries, forestry and livestock) (Van Fish, Van Trade, Sol Trade, Sol Agri). They also identified explicit concerns for the economic vulnerability of farming and fishing communities (Sol Fish, Van Agri, Sol Agri). Agricultural policy expressed concerns that poor resource management would additionally increase vulnerability of communities to climate change (Sol Agri).

Trade-related policy retained an economic framing regarding the consequences of environmental aspects of food systems, commercial exploitation of natural resources by export industries (specifically fisheries, forestry and livestock) (Van Fish, Van Trade, Sol Trade, SI Ag) and other associated negative environmental practices. Concerns around the overexploitation of natural resources were presented as a key risk for export industries and to the sustainable development of trade (Sol Trade, Van Trade).

*"Positive measures need to be taken to ensure that the environment is not harmed. This might involve the regulation of exploitation of natural resources central to key export industries, such as forestry, fisheries, and mining. A failure to regulate may harm other export industries, such as tourism, or prevent eco-certification that may be important for the maximisation of the return from some industries."* Sol Trade

Environmental exploitation and degradation were attributed to historically poor execution of environmental assessments (particularly in fisheries and forestry through trade) (Van Trade), inadequate management of fish stock (Sol Fish) and poor environmental practices that were being exacerbated by lack of knowledge by farmers on animal and farming management strategies to reduce environmental degradation (e.g., fallow periods, soil fertility, livestock) (Sol Agri, Van Agri, Van Livestock). According to the Vanuatu Agriculture Sector Policy (2015–2030), climate adaptation projects had not yet been able to address deficits in climate knowledge, information and technology for the most remote or vulnerable farmers.

### 3.2.3. Key Concern 3: Achieving Food and Nutrition Security Is a Concern for Food Systems Sectors

Productive sector policies (agriculture, fisheries and livestock) in both countries framed food security and nutrition as issues that provided them with a sense of purpose (Table 4). For example, the Solomon Islands National Fisheries Policy (2019–2029) identified fish as critical to balanced diets and NCD alleviation, with key strategies including the safeguarding of inland and inshore fisheries and establishing aquaculture programmes that meet future population protein requirements. The overall vision of the Vanuatu National Fisheries Sector Policy (2016–2031) is a "Healthy and sustainable fisheries sector for the long-term economic, social and food security for current and future generations". The Vanuatu National Livestock Policy (2015) regards livestock as having the potential to be a major contributor to food security.

Food security and nutrition policy concerns were articulated as a growing shortfall between supply and demand in coastal fisheries (Sol Fish), resulting from poor fisheries production (Van Fish) and reduced per-capita production of food crops by subsistence farmers (Sol Agri). Production gaps were thought to be exacerbated by population growth, and the sense of uncertainty regarding the magnitude of yields at any one time. For example, the Vanuatu Agriculture Sector Policy (2015–2030) reported that declining and inconsistent production was undermining food availability. The Solomon Islands Agriculture Sector Growth and Investment Plan (2021–2030) associated poor sectoral performance to a declining contribution to food security, which had apparently escalated during the COVID-19 pandemic. The Solomon Islands National Fisheries Policy (2019–2029) hypothesised that shortfalls in fish production required to maintain food security would reach 4000 tons by 2030.

> *"Calculations suggest coastal fisheries will not supply the fish required for future food security, with projected shortfalls of more than 4000 tonnes per year in fish supply versus demand by 2030"*

Food produced by the livestock sector was given as an alternative and relatively cheap source of protein, with potential to address future gaps in fisheries outputs (Sol Agri).

### 3.3. Policy Aims and Instruments

3.3.1. Aim 1: To Increase Food Production for Import Substitution and Export Trade, and to Develop Industry

Food systems policy aims had a strong productivist and economic focus, including improving production, and maximising economic gains (Table 4). In particular, trade and agricultural policy aimed to increase the contribution of productive sectors to trade, while trade, agriculture and industry development policies all promoted domestic value-adding. In general, production aims were never explicitly in fulfillment of food security purposes. For example, both the Solomon Islands Agriculture Sector Growth and Investment Plan (2021–2030) and Vanuatu Industry Plan established goals to promote substitution across a range of specific food imports, and both identified niche and specialty crops for export.

Authoritative instruments applied to achieve economic aims included those that establish a transparent set of rules for commercial engagement, such as an Industry Development Act (2014) in Vanuatu, a Consumer Protection Bill (1995) and a Measurements and Weights Act (1996) in the Solomon Islands, and Price Controls Acts in both settings (Table 5). The Trade Policy Framework (2015) identified food-related legislation as critical to maintaining coherence with preferential trade arrangements. Authoritative tools were also applied to facilitate land allocation and industrial zoning specifically for agriculture, intensive farming, and the fish processing industry.

Authoritative instruments were also used to promote foreign investment. Vanuatu's Foreign Investment Act (2019) enables foreign investors to engage in any investment activity and offers foreigners treatment equal to nationals in the establishment, expansion and operation of investment. Vanuatu's trade and investment policies specifically recommended that investors be offered unfettered and non-discriminatory access to rural land for productive purposes.

In both countries, new regulations were proposed to improve quota management and payment in both countries, including tightening fisheries licensing schemes. Trade policy recommended regulating fisheries management conditions differentially for small scale fishers, and protections for 'infant industry' in trade policy space (Vanuatu Industry, Van Trade). A Protected Industries Act (1996) in the Solomon Islands was adopted to restrict imports of products that may impair local industry development, while Vanuatu's Industry Development Act (2014) introduced the possibility of imposing taxes on the export of products that have not undergone value addition.

> *"Subject to this Act, any investment activity may be carried out by a foreign investor in Vanuatu, unless the investment activity is a prohibited activity- no restrictions are placed on foreign investor participation in onshore fish processing or ancillary services, or in livestock sector"* Van Trade

Food systems policies employed a number of direct incentives to induce or extinguish actions with the aim of promoting production and economic opportunity. Direct incentives were mostly applied in fulfillment of the aim of expanding primary production capacity, encouraging greater efficiency, and maximising value-adding and processing opportunities. For example, direct incentives were applied in the Solomon Islands Agriculture Sector Growth and Investment Plan (2021–2030) to boost livestock and apiary industries, with the provision of roosters, boars, mini feed mills, tools, and hives. Incentives were also applied to aid value adding and production, including the provision and installation of copra driers, and the introduction of technologies and mechanisation for the tilling, crushing, and processing of coconut products (Sol Ag). Direct incentives included taxes on the

export of primary products that negated value-addition (Van Industry), grant or incubator funding, or co-investment opportunities. Incubators were offered to both local and foreign entrepreneurs to accelerate industrial development.

**Table 5.** Summary of policy instruments used to address food systems objectives.

| | Policy Aims | | |
|---|---|---|---|
| | **Increase the Contribution of Productive Sectors for Import Substitution and Export Trade** | **Promote Environmentally Resilient Food Supply** | **Produce Enough Food to Meet Population Requirements for Healthy Diets** |
| Authoritative instrument | • Regulations to promote direct foreign investment and trade<br>• Regulations to stem export of non-value-added goods<br>• Regulations for commercial engagement<br>• Fisheries protections<br>• Industry development regulations | • Fisheries protections<br>• Environmental impact assessments<br>• Biosecurity laws to prevent animal and plant disease | • Food safety regulations<br>• Price controls |
| Incentives | • Inputs to farming and fishing<br>• Equipment for value adding and manufacturing<br>• Technical and advisory services<br>• Backward linkage programmes to connect suppliers to local procurement opportunities<br>• Collaboration between farmers, vendors, middlemen and others across the value chain<br>• General infrastructure critical to market access<br>• Subsidies for production of export goods<br>• Land use planning | • Waste technology improvements and infrastructure<br>• Technological enhancements for sustainable farming | • Fiscal policies to promote healthy consumption<br>• Inputs to farming<br>• Improvements to market infrastructure |
| Knowledge and skills building | • Capacity building of private sector and entrepreneurs on high value production, processing and marketing<br>• Business development support and training | • Research, development, testing of sustainable production methods and climate resilient varieties<br>• Training and promotion of best-practice for sustainable farming and fishing<br>• Fisheries information systems | • Food and nutrition guidelines and databases<br>• Establishment of a food council |

Indirect incentives were largely applied to achieve economic aims in ways that would overcome factors with potential to undermine production, business development and market participation. Indirect incentives usually took the form of services, infrastructure, subsidies, protection zoning, and market and partnership linkages. Services provided

required technical and advisory expertise not commonly available to primary producers and small businesses, for instance, on-site soil analysis improvement consultations, animal health laboratories and veterinary services, business development centres and industrial parks, programmes to reduce complexity in the export of manufactured products ('Seamless Trade'), equipment measuring and calibration, and machinery and technology centres offering maintenance, standards certification, biosecurity surveillance and pest eradication. Some incentives were stipulated as a mechanism to facilitate the entry of women and youth entrepreneurs into small-to-medium enterprise, but strict quotas were not evident. Vanuatu was establishing an Import Substitution and Export Finance Facility to manage low interest loans that reduce startup expenses as a barrier to business establishment.

Indirect incentives were also used to strengthen market access, by building productive relationships that might improve coordination and linkages between producers, processors and traders. For instance, the establishment and strengthening of farming and livestock associations and cooperatives (particularly for livestock and cocoa operations), and the facilitation of new public–private partnerships across the food chain. As an incentive to maximise production capacity, greater leveraging of public procurement was also introduced. Both trade frameworks proposed scaling up distribution linkages with local suppliers (hotels, restaurant, retailers, public procurement, wholesalers) to facilitate market access and improve reliability and safety of supply for export (Sol Trade, Van Trade). The Solomon Islands Agriculture Sector Growth and Investment Plan (2021–2030) specified Export and Import Substitution Targets, framed as commodities with main export potential (copra, crude coconut, coconut oil, cocoa means, kava) and commodities for import substitution (poultry, eggs, pork, beef, rice).

Additionally evident were tax exemptions and subsidies offering incentives for participation in the trade and export market. For instance, the Solomon Islands Government subsidises the copra, noni and cocoa industries for export to counter high costs of freight. The Solomon Islands was considering offering GST relief, duty exemptions, and income tax reductions that attract direct foreign investment, and a particularly generous "tax holiday" was offered in Vanuatu for agricultural investments. Tax exemptions and tariffs were also applied in Vanuatu to promote value addition prior to export (on the basis of value-add criteria) (Van Industry), and both countries offered reduced import excises to promote production and manufacturing (agricultural and fishing inputs, product packaging).

Food systems policy documents also committed to providing infrastructure that would improve market access, for instance, slaughter facilities, quality market storage facilities (Sol Agri, Van Agri), shared machinery centres, and the commitments for greater agricultural land allocation (Sol Planning). In addition to this, both countries had dedicated strategies for improving transport, communications, waste and water infrastructure, to reduce electricity tariffs, provide reliable access to clean water, roads and transport infrastructure, and to provide populations with improved digital infrastructure. Though distal to food systems, this infrastructure is critical for maximising participation in local and regional markets by providing opportunities for communication and coordination across value-chains, and paving reliable/affordable access to markets.

Knowledge and skills building instruments were evident in policy documents promoting increased production and to facilitate industry and business development for small to medium enterprises. Research was applied to build knowledge around appropriate livestock breeds (Sol Agri), and exotic and high value fruit and vegetables production. Research was also used to identify business opportunities, for instance in aquaculture breeding (Van Fish), or opportunities for primary product processing (Sol MCIL). Both countries committed to building knowledge and skills in farm establishment, and farm mechanisation to lift the scale of production for farmers, fishers, and producer groups (Sol Agri, Van Agri, Van Fish). The Solomon Islands planned to expand transport and communication capability across extension services to enhance communications (Sol Agri). The Solomon Islands was committing to undertake supply chain mapping for key commodities to improve tracking and capacity in handling, processing, and packaging for cold

chain (Sol Agri). In Vanuatu, there were plans to adopt new market information systems to facilitate reliable, informed access to markets (Van Agri), and scale up the role of Vanuatu's Cooperatives Business network to coordinate supply and demand (Van Indus).

Both countries were applying knowledge and skill-building instruments to improve business opportunities, including improving financial literacy of producers and small businesses, with training and coaching to assist with gaining access to credit (Sol Industry, Van Agri), and with business training and support via 'incubators' (Sol Industry) and 'business centres'. Vanuatu had plans to create a series of knowledge products to aid entry of small businesses into business startup, identify market opportunities, product development, standards, branding, and marketing (Van Industry). Both countries committed to building knowledge and skills of producers and entrepreneurs in high-value food value-adding, marketing and export (Sol Trade, Van Industry), which, in the Solomon Islands, included coffee, nuts, dried fruit and banana chips, ginger and other spices, and cassava chips and flours (SI trade), and in Vanuatu, crustacean processing, baked goods, jam products, vanilla, nuts, spices, cereals, ice creams, processed vegetable products, dates, figs, avocadoes, legumes, exotic juices and a range of coconut-based food items.

### 3.3.2. Aim 2: To Promote an Environmentally Resilient Food Supply

Food system policy documents reflected aims to better manage natural resources and promote resilience (Table 4). In particular, fisheries and trade policies in both countries included priorities to promote the sustainable management of coastal and fisheries resources, and to prevent resource exploitation (Sol Fish, Van Fish, Sol Trade, Van Trade). In comparison, agricultural sector policy in both countries included a number of aims to promote resilience and disaster preparedness.

Authoritative tools were applied to mitigate overfishing and illegal fishing (Sol Fish, Van Fish) and to reducing reef endangerment (Table 5). For instance, restrictive instruments were in place in Vanuatu that banned the commercialisation of endangered fish sources, enforced importation of destructive fishing gear, and applied fish product traceability systems (Fisheries Management Act, 2014). Vanuatu also offered protection to specific areas to prohibit them from aquaculture development or related activities. Authoritative tools were also used to achieve aims to reduce environmental exploitation, for instance, environmental protection acts were employed in both countries to prevent degradation through practical means (e.g., waste control, recycling). Environmental impact assessments were mandatory in new developments, though trade policies alluded to these being poorly implemented or enforced.

In comparison, policy aims to promote an environmentally resilient food supply were largely addressed with incentives and knowledge and skills building instruments. Incentives included technological enhancements that promote sustainable farming practices (Sol Agri), taxes to discourage the import of plastics, and the scale up waste and composting infrastructure (Sol Infrastructure, Sol Agri). Knowledge and skills transfer instruments, such as research and development programmes, included technical advice and extension services, training and knowledge resources. Both countries committed to research and knowledge transfer around climate and pest resilient crops and soil improvement strategies, and promotional activities around sustainable and organic farming practices. In fisheries policy, knowledge and skills building activities included fisheries information and management systems, research on fisheries repletion and invasive species, and community education.

*"Promote organic farming through awareness, training and certification"* Van Ag

*"Strengthen traditional and self-reliant agricultural systems through development and implementation of programs with components that encourage growing traditional climate-resilient staple crops such as sweet potato, taro, banana, yam, cassava and trees and animals"* Van Ag

### 3.3.3. Aim 3: Produce Enough Food to Meet Population Requirements for Healthy Diets

A number of food systems policy documents represented as key sectoral objectives to promote food security and nutrition (Table 4). For example, the main goal of Vanuatu's National Fisheries Sector Policy (2016–2031) was to "increase production, people's incomes, food security, and nutritional status of all ni-Vanuatu", while Vanuatu's Trade Policy Framework Update (2019–2025) called for development of the livestock sector to promote food security. The Solomon Islands Agriculture Sector Growth and Investment Plan (2021–2030) included in its mission to enhance food security for all rural and urban areas.

However, compared with other policy priorities, nutrition and food security aims were addressed with a more limited range of policy instruments (Table 5). In both countries, authoritative tools were in place to maintain food safety, and price controls were applied to prevent price fluctuation for food security and equity purposes. Though, both less healthy products (e.g., corned beef, sweetened condensed milk, biscuits and sugar) and more healthy foods (tinned fish, rice, oil and flour) were protected by price controls. Both countries trade frameworks referenced the need to protect health, reiterating the importance of health and safety regulations and addressing NCDs.

Direct incentives applied to achieve food and nutrition security aims included sugar-sweetened beverage taxes in the Solomons (Sol Budget), the zoning of marine protected areas to protect community food reserves (Van fish), and the provision of planting material for high nutritional value food crops (Van Agri, Sol Agri). The Vanuatu National Fisheries Sector Policy 2016–2031 committed to upgrading market infrastructure and market access at fish markets (Van Fish).

Knowledge and skills building instruments applied to achieve food and nutrition aims largely centred around knowledge and awareness raising activities for consumers and farmers on more nutritious food alternatives and through research. The agricultural strategy of both countries recommended that local foods and healthy balanced diets be promoted using knowledge and awareness raising activities, in partnership with health departments. In the Solomons, research was planned to document traditional practices of production and preservation, identify nutrient dense species, and to determine feasible technological enhancements for harvest and post-harvest food handling and storage of fruit and vegetables, for example hydroponics and seaweed fertilization (Sol Agri).

## 4. Discussion

This study used policy content analysis to study the frames employed by different food systems sectors, providing critical insight into sectoral understandings of policy concerns and priorities [51,52]. We identified a predominately economic framing of issues affecting food systems sectors, and clear policy aims to increase the contribution of productive sectors to export trade and import substitution. Producing enough food to meet population requirements for healthy diets was presented as giving purpose to the productive sectors. Food systems sectors also had clear aims to promote environmentally resilient food supply and mitigate future impacts of environmental and natural resources degradation on food productivity. Through our instrumentation analysis, we found that policy concerns most strongly operationalised in food systems policy were those that increased production quantities at the farm level, promoted value-adding and business development opportunities, or regulated the management of ocean resources. In contrast, concerns around nutrition, healthy diets and environmental resilience were only minimally addressed, with policy instruments tending to rely on knowledge and skills building for producers and consumers. Tensions in aims for cross-sectoral food policy are evident in other settings [53,54]. We explore these below, outlining further opportunities for food systems policy to simultaneously promote positive economic, environmental and nutrition outcomes.

Overall, we found that concerns of food systems policy sectors predominately revolve around their economic contribution. While an economic framing is an important aspect of food systems, and consistent with global food system goals to utilise food systems as an opportunity to contribute to economic recovery following the COVID-19 pandemic [13],

our findings suggests that food is predominately viewed by governments as an economic resource [53,55,56]. In the Solomon Islands and Vanuatu, declining food production and import dependance were most often regarded as a failing of farmers and fishers to meet production quantities sufficient for export demand and local consumption requirements. As in other contexts, food import reliance in the Pacific has developed over decades in response to the dietary transition and ongoing urbanisation increasing the availability, affordability, and visibility of unhealthy processed foods in LMICs [2,57–60]. By applying an instrumentation lens, we were able to demonstrate how the prioritisation by food systems sectors of production for economic growth and resilience is operationalised. In particular, this has been conducted through the number of more 'coercive' policies employed to promote trade, fisheries reporting, foreign investment, price controls, food standards, and through the broad range of incentive-based approaches targeting different facets of food production and market access. This builds on previous work focusing on policy framing to examine variations in sectoral objectives in relation to food [61–63].

Our findings, regarding a focus on increased production as a goal in itself and as the main pathway to achieving food security, suggests a 'productivist' approach to food policy that focuses on addressing food security by increasing production [64,65]. Consistent with findings elsewhere, this focus seems to have led to an emphasis on production gaps as hampering efforts to achieve food security, and the patchy policy attention to 'downstream' aspects of food systems such as market access, transport, storage and marketing [56,66]. However, the instrumentation analysis also indicated the integration of food security and nutrition considerations in trade and industry policies, which extends beyond a narrow productivism policy paradigm. As such, this approach to food policy analysis can provide additional nuance to understandings of how policy is operationalised across sectors and, thus, the paradigms at play.

While food systems policy actors clearly hold concerns for the impact of climate change and natural resource exploitation on food production, we found a linking of environmental impacts to economic impacts, particularly by trade and financial sectors. Another critical finding was that environmental concerns were only minimally addressed, with a combination of knowledge and skills building strategies promoting sustainable practices at the farm, and authoritative instruments (e.g., environmental protections and quota management) that were apparently poorly executed due to enforcement capacity. Globally, most approaches to improve environmental practices during food production target the farmer or the fisher rather than the whole food system [67]. However, promoting environmentally resilient food systems are likely to need backing by a stronger set of instruments enforcing and incentivize structural changes across the whole food chain [68,69].

Similarly, the production of food that meets population requirements for healthy diets was a key mission for productive sectors in Solomon Islands and Vanuatu. These aims were operationalized mainly through knowledge and skills strategies to promote healthier crops and healthy consumption behaviors, apart from legislation regarding food safety and quality for export. Averting nutrition challenges by incentivizing, regulating and promoting nutrition-sensitive food systems is more efficient than treating nutritional conditions [55]. However, this paper has found that promoting nutrition-sensitive food systems are not key yet policy priorities of food systems sectors, resonating with concerns that the promotion of healthy diets as 'Health's responsibility' [15,16].

A final reflection from this analysis is that food systems policies demonstrated that policymakers in both countries were engaged in policy-oriented learning [47], in that they were acknowledging some of the food systems challenges and moving forward to address them. For instance, technical and mechanical services were offered to overcome shortcomings in farm production, and market storage was offered to address waste issues. Additionally, we noted similarities in food systems policy concerns and approaches across both countries, suggesting regional policy coherence and transnational learning [70].

### 4.1. Opportunities for Win-Win-Win Food System Policy

This research has confirmed that efforts to increase action on nutrition and environmental sustainability in food systems policy will need to be cognizant of the priority for economic development by food systems sectors, and the differences in cross-sectoral objectives across food systems sectors. Key to this will be the identification of positive synergies across all dimensions of healthy and sustainable food system policy despite the divergent set of specific concerns and priorities [71,72]. In fulfillment of our aim, we outline below a series of policy 'win-win-wins' that are nuanced to country context.

We found that food systems policies in the Solomon Islands and Vanuatu reflected a desire to create greater efficiencies across the food system, however, there is opportunity for policy efforts to center more closely on production, distribution and marketing of healthy plant-based foods and proteins for consumption as opposed to export [11]. Scaling up production, manufacture and marketing of healthier foods is likely to require stronger planning and cooperation across the value chain for foods with the most potential to lead to desired food-systems outcomes [73]. Dietary patterns that are likely to have a reduced environmental impact include those that are plant-based, and replace ruminants with other protein sources, including fish, poultry and pork [67]. To support this shift, national food-based dietary guidelines could be reoriented to supporting a healthy sustainable food supply [11], which the Pacific Islands Guide for Healthy Living already does well through its promotion of local crops and proteins [74], providing a strong basis for food systems. Given the experience elsewhere of an orientation to cash-crops and animal protein, this opportunity is likely to have global resonance [75].

We found that both countries had in place a comprehensive set of policy instruments for boosting food production opportunities, for instance, by funding aquaculture innovations, or offering direct incentives for livestock and crop production. However, we also identified opportunities to extend food system policy efforts beyond production to more fully address other downstream aspects of food systems, including transport, manufacturing and markets [36]. Food systems policies in Vanuatu and the Solomon Islands noted value chain mapping as a potential policy approach, and though value chain mapping rarely incorporates both environmental and nutrition dimensions [76], there is potential for both countries to reorientate value chain approaches to foods with both positive nutritional and ecological outcomes [77–79]. Examples of this include preferencing support with accessing credit and inputs to producers contributing to a nutrition-sensitive food system, and encouraging operators across the supply chain operators to better uptake technological solutions for communication, contracting and logistics [73]. Though there are plans to address key aspects of value chain (including market forecasting and planning) for key export produce (e.g., organic foods and pressed juices), efforts to create timely market information and facilitate the reliable and efficient passage of food across the food chain could better emphasise specific foods and consumption patterns required for national food and nutrition requirements [1,80]. For example, supporting entrepreneurship in fermentation methods of food processing taps into multiple policy aims of food systems sectors, including value-addition and entrepreneurship, nutrition and cultural values, vulnerability and ecosystem protection [81].

Smallholder producers and small to medium enterprises (SMEs) in the Pacific clearly face a number of barriers to capitalising on policy incentives, to building knowledge of market opportunities, and in maintaining reliable market access for their products [82]. Enhancing competitiveness for SMEs is already a key aim of the sector, for example, by promoting value-adding and improving their marketing access. However, there is opportunity to better support smallholders (globally, responsible for over one-third of the world's food supply [83]), and SMEs with product portfolios focused on nutritious and plant-rich foods for national consumption, thus, contributing to aims around import substitution. Structural drivers for this include improvements to rural infrastructure, support with accessing innovative technologies and improved methods, and through activities that support commercialisation [83]. Policymakers can also facilitate 'Coopetition'

(the act of cooperating and competing) in the agrivalue chain to build proficiencies in the sharing and pooling of competencies [73,84]. They can then orientate 'coopetition' so that it achieves policy priorities across all dimensions of food systems, extending to those around inclusion, resilience and gender [73]. The development and ongoing support of SMEs offers great opportunity for strengthening healthy and sustainable diets [82,85]. Value chains can also be exercised and strengthened by harnessing the power of public and private purchasing by fostering 'backward linkages' [86] through public procurement, and by harnessing fisheries processing industries in both countries, tourism in Vanuatu and mining in the Solomon Islands.

Our analysis also noted opportunities to integrate nutrition and environmental concerns into some of the existing authoritative instruments. For instance, fiscal policies could alter incentives across the value-chain to boost efficiency and value-adding of foods with health and ecological benefits [80]. Countries could extend sweetened beverage taxes towards foods with a low nutritional value or with clear environmental impacts [11]. Environmental impact assessments have demonstrated benefits associated with following dietary guidelines [87], and in the Solomon Islands and Vanuatu environmental impact assessments could be expanded to food [88,89], and capacity to enforce them improved so that public and private sector are forced to align with food systems aims. Theoretically, trade and investment policy should be reoriented to support nutritional and environmental aspects of food systems [90], though Pacific Island countries have not traditionally had a strong voice at the negotiation table [91]. Traditionally, food security and nutrition advocates have failed to translate their intentions and beliefs to policy outside of the health sector [92,93], and have lacked the resources and power to engage with political leaders effectively [53,94]. Sustainability concerns have been managed in a similarly superficial way [67].

### 4.2. Policy Coherence for Promoting Healthy and Sustainable Diets in Food Systems

We have demonstrated clear differences in core policy aims of different food systems sectors, a factor that is a known barrier to effective and sustained multisectoral policy action [16,94,95]. The promotion of policy coherence across different aspects of development is a target of the sustainable development agenda [72], and many countries are working towards this already. Both countries, through their participation in the UN Food Systems Summit and hosting of national food system dialogues, have demonstrated interest and commitment to the transformation of food systems that can be harnessed to better address health and environmental dimensions. But nutrition issues suffer from being an invisible and slow-burn issue [94], and climate change approaches in the Pacific have been historically managed with siloed approaches [96].

This study also points to avenues to more effectively frame food systems 'transformation' to improve outcomes for nutrition and the environment, such that it becomes a domestic policy priority [13,42,51,53]. One key framing will be that food systems have potential to simultaneously address multiple sustainable development objectives extending beyond those explored in this paper, including human rights, gender equity and youth development [11,97]. A framing that taps into the economic priority of both countries would be to more clearly identify and quantify negative externalities associated with food production [98,99]. Academia has a role to play by contributing the evidence-base needed for the prioritisation of food systems strengthening and for efficient policymaking [100], for instance, by internalising the costs of a product's effects on the environment and human health into its price [11]. Academia could also work to highlight the political and social determinants of policy [11] and the degree to which policy change is occurring and having impact.

## 5. Conclusions

This study has identified clear differences in the core policy aims of different food system sectors that may be present elsewhere. Previous research on policy coherence within food system sectors suggests that these findings reflect the situation in other LMICs [101], as well as in HICs [102,103]. While providing for the nutrition requirements of the population, and promoting environmentally resilient food systems, are given as core aims of food systems sectors, food is predominately considered by government as an 'economic good' with a key contribution to economic development. In recognition of this, policy advocates for healthy and sustainable food systems will need to advocate that food system policies better emphasise the specific foods and consumption patterns required to meet national food and nutrition requirements, and not just those with export potential.

In order to realise the transformation of food systems, the integration of nutritional and environmental concerns is likely to require that countries adopt a stronger set of instruments that enforce and incentivise structural changes across the whole food chain. High level cross-ministerial political leaders will be needed to oversee the coordination and implementation of the multisectoral agenda, offering incentives, oversight and support. This points to the need to more effectively frame food systems 'transformation', such that it becomes a domestic policy priority for countries

**Author Contributions:** E.R. drafted the manuscript; E.R. and A.-M.T. were involved in all aspects of the study. A.R. and A.-M.T. provided supervision and review to the manuscript. S.M., D.W., A.F. and E.J. supported data collection and analysis and undertook technical review of the manuscript. All authors have read and agreed to the published version of the manuscript.

**Funding:** E.R., A.-M.T., E.J. and D.W. are supported by ACIAR Pacific Food Systems project FIS/2018/155. A.F. is supported by the University of Wollongong and ACIAR Pacific Food Systems project FIS/2018/155. Senoveva Mauli is supported by the ACIAR funded Pathway Project titled "Strengthening and scaling community-based approached to Pacific coastal fisheries management in support of the New Song".

**Data Availability Statement:** Not applicable.

**Acknowledgments:** The authors would like to thank members of the Pacific Food Policy Project Advisory Group for their input and direction.

**Conflicts of Interest:** The authors declare no conflict of interest.

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
