# Peer review of "Strengthening Food Systems Governance to Achieve Multiple Objectives: A Comparative Instrumentation Analysis of Food Systems Policies in Vanuatu and the Solomon Islands"

_sustainability, doi:10.3390/su14106139_

Round 1
Reviewer 1 Report
Review the format of the tables and adapt it to the MDPI format.
In conclusion, try to extrapolate the results to other parts of the world with characteristics similar to Vanuatu and the Solomon Islands.
Reviewer 2 Report
Line 78.
I like to read about the area o territories that are involucrated with the global food supply. Data or specific information of Pacific Islands Countries impacts.
Does it have any link with Sustainable Development Objectives Agenda? Please explain.
Line 94
Specify the range of analysis years. Some policies may be older but are current.
Indicate the type of methodology you are using.
Line 170
I can not appreciate well the INVIVO analysis results.
The qualitative analysis should be clearly expressed in this section.
Line 178
Table 3 may be in the Methodology section.
Line 506
Are these policy instruments working together?
How does INVIVO interlink the information to compare Vanuatu and the Salomon Islands?
Do you have results of your variables - "codes"?
How do you measure levels of governance and then the multiple objectives?

Reviewer 3 Report
The manuscript entitled “Strengthening food systems governance to achieve multiple objectives: a comparative instrumentation analysis of food systems policies in Vanuatu and the Solomon Islands” presents interesting topic, however I do not see any need for publication such document in international journal. The international context must be emphasized. What can we learn from this study?
- The abstract should be a single paragraph and should follow the style of structured abstracts, but without headings (see authors guideline)
- In this section Authors presented the information associated with food systems policy (mostly only in Vanuatu and the Solomon Islands). This section should be improved a little bit – what do we know and what is the background for this study. Some detailed information about other studies are necessary (international context – the situation in other countries should be presented). The good background should present the history of problem, the current knowledge and scientific "gap", and then authors should present how their study could fill this gap to justify the study.
- Table 1 – for ‘Infrastructure’ (Instrument typology) – please add the reference
- Line 139 – DATA – please provide more detailed information, e.g. ‘We used government organisational structures’ – how you identified it?
- Line 149 – ‘expert advisory group’ - please provide more detailed information about experts
- Table 4 – there are black and white dots - it is intentional? If so, please add
Minor comments
- Line 563 – there is some typos –‘ system[66]. However,’
Round 2
Reviewer 2 Report
Observations were taken satisfactorily.

Reviewer 3 Report
I appreciate the great efforts that the authors have made in response to my questions and concerns. I have no further comment